# YOU ONLY LOOK AT SCREENS: MULTIMODAL CHAIN-OF-ACTION AGENTS

## ABSTRACT

Autonomous user interface (UI) agents aim to facilitate task automation by interacting with the user interface without manual intervention. Recent studies have investigated eliciting the capabilities of large language models (LLMs) for effective engagement in diverse environments. To align with the input-output requirement of LLMs, existing approaches are developed under a sandbox setting where they rely on external tools and application-specific APIs to parse the environment into textual elements and interpret the predicted actions. Consequently, those approaches often grapple with inference inefficiency and error propagation risks. To mitigate the challenges, we introduce Auto-UI, a multimodal solution that directly interacts with the interface, bypassing the need for environment parsing or reliance on application-dependent APIs. Moreover, we propose a chain-of-action technique—leveraging a series of intermediate previous action histories and future action plans—to help the agent decide what action to execute. We evaluate our approach on a new device-control benchmark AITW with $30K$ unique instructions, spanning multi-step tasks such as application operation, web searching, and web shopping. Experimental results show that Auto-UI achieves state-of-the-art performance with an action type prediction accuracy of 90% and an overall action success rate of 74%. Code is publicly available at `Anonymous`.

## 1 INTRODUCTION

Building intelligent autonomous agents that are capable of task planning, decision making, and action execution in a particular environment is a long-standing goal of artificial intelligence (AI) (Searle, 1969; Wooldridge & Jennings, 1995; Maes, 1995; Hendler, 1999). The advent of large language models (LLMs) (Brown et al., 2020; Chowdhery et al., 2022; OpenAI, 2023) has flourished promising opportunities for developing autonomous agents to assist users in completing tasks in distinct environments such as operation systems, specific applications, and web browsers (Adept, 2022; Rawles et al., 2023; Liu et al., 2023; Zhou et al., 2023; Wang et al., 2023c).

Recent studies have explored prompt engineering (Richards, 2023; Nakajima, 2023; Reworkd, 2023; Sumers et al., 2023; Liu et al., 2023) and fine-tuning techniques (Rawles et al., 2023; Wen et al., 2023; Sun et al., 2022) to elicit the capability of language models to execute actions in interactive environments. However, there are at least two major challenges that have limited real-world applications of autonomous agents.

First, existing approaches commonly rely on external tools such as optical character recognition (OCR) and icon detectors (Zhang et al., 2021; Sunkara et al., 2022) to parse the environment into textual elements (e.g., HTML layouts) as inputs to a language model (Figure 1(a)) (Rawles et al., 2023; Wen et al., 2023). On the one hand, the parsed elements generate lengthy inputs, thus leading to inference inefficiency. Since computational latency is a key measure in deployment, using lengthy inputs would increase inference cost and may even exceed the input length limit of the language model. On the other hand, parsing the visual environment into textual elements may also be prone to error propagation or information loss because parsing mistakes are inevitable using external tools.

Second, most existing approaches are under the sand-box setting that requires accessing internal APIs to interact with the environment (Zhou et al., 2023; Gur et al., 2023), e.g., using a JavaScript element selection on a webpage or a Python interpreter to execute actions. However in practice, the API interface is often inaccessible in third-party applications (Apps).

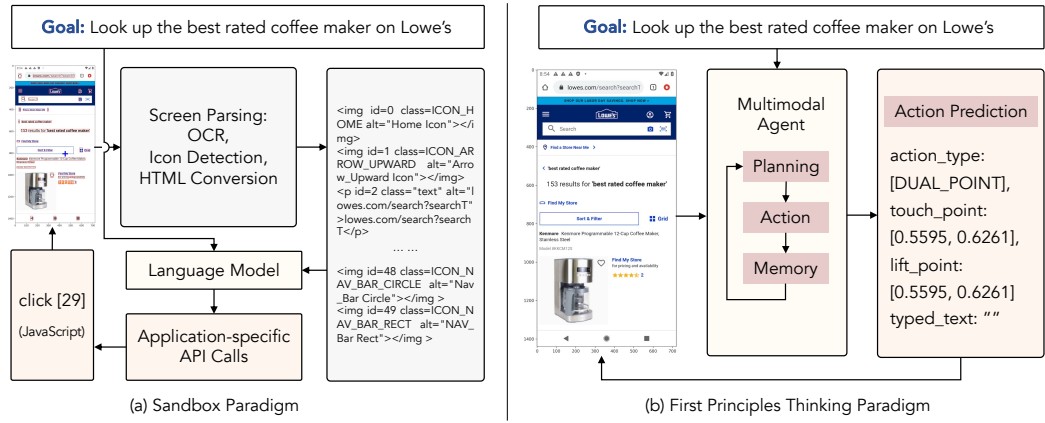

Figure 1: Comparison of two autonomous agent paradigms. The sandbox paradigm depends on the intermediate transformation between environments and agents, i.e., needing access to intermediate environment parsing or interval application-dependent APIs. In contrast, our first principles thinking paradigm allows direct interactions on the screen without intermediate transformation. Details of the action types and action points are presented in Section 3.3.

These challenges have motivated more advanced techniques that are capable of *first principles thinking* (Aristotle; Irwin, 1989)—allowing direct interactions on the screen without needing access to intermediate environment parsing or interval application-dependent APIs (Figure 1(b)). To address the challenges, we introduce **Auto-UI**, a multimodal approach that directly interacts with the interface. To improve the agent's action prediction capability, we propose a novel **chain-of-action** technique, where a chain of action is a series of intermediate previous action histories and future action plans that lead to action prediction.

We evaluate Auto-UI on a new device-control benchmark AITW (Rawles et al., 2023) with $30K$ unique instructions, spanning multi-step tasks of application operation, web searching, and web shopping. Experimental results show that Auto-UI achieves state-of-the-art performance with an action type prediction accuracy of 90% and an action success rate of 74%.

In summary, our work makes the following technical contributions:

(i) We introduce Auto-UI, a multimodal agent for autonomous UI control that can directly interact with the screens, thus circumventing the constraints of environment parsing and application-specific API access.

(ii) We propose a chain-of-action technique that leverages the previously executed actions and future action plans to help the agent decide what action to execute at each step.

(iii) Auto-UI achieves state-of-the-art performance with an action type prediction accuracy of 90% and an action success rate of 74%. Notably, Auto-UI can infer an action as fast as within less than one second.

## 2 RELATED WORK

Our work falls into the field of language agents. This section will first review the recent progress in building language agents and then discuss the approaches to conduct user interface control with language agents.

### 2.1 LANGUAGE AGENTS

Language agents refer to those agents that can follow user instructions and interact with environments to complete tasks. Such agents expand the landscape of language models to compete in specific fields, including application operation, web searching, and web shopping. There are two popular

types of language agents, autonomous agents and communicative agents. Autonomous agents aim to assist humans to achieve specific goals in the real world. Typical examples of autonomous agents are AutoGPT (Richards, 2023), BabyAGI (Nakajima, 2023), and AgentGPT (Reworkd, 2023). In contrast, communicative agents are personalized and socialized agents (Park et al., 2023; Wang et al., 2023b; Zhu et al., 2023; Hong et al., 2023) with human behaviors that can communicate and collaborate with each other. They are often deployed in immersive environments. Inspired by the potential in real-world applications, this work focuses on autonomous agents, especially those working in mobile devices. We aim to assist users by completing multi-step tasks (e.g., manipulating Apps, web shopping, and question answering) without any manual intervention. Given a user instruction in natural language, the agent is required to interpret the instruction and execute actions by directly controlling its user interface. Due to the requirement in real-world applications, the agent is expected to be both effective and efficient.

## 2.2 UI CONTROL WITH NATURAL LANGUAGE

Recently, LLMs have shown promise in building autonomous UI agents with abilities of instruction following (Sanh et al., 2021; Taori et al., 2023b; Chiang et al., 2023) and chain-of-thought (CoT) prompting (Nye et al., 2022; Wei et al., 2022). Especially, CoT prompting (Wei et al., 2022; Kojima et al., 2022; Zhang et al., 2023a) elicit LLMs' capacities of step-by-step planning, decision making, and action execution. Those capacities have been shown to be effective in UI control tasks (Rawles et al., 2023). However, the task environments are graphical user interfaces (GUIs), instead of natural language that LLMs can directly process. Therefore, the GUI states and actions are required to be converted to textual formats to conform to the input and output formats of LLMs. For example, it is feasible to parse the UI screens by icon recognition and OCR (Zhang et al., 2021; Sunkara et al., 2022) and organize the parsed elements into HTML layouts. As a compromise, existing approaches are restricted in a sandbox setting where they rely on external tools (Rawles et al., 2023; Wen et al., 2023) and application-specific APIs (Zhou et al., 2023; Gur et al., 2023) for environment parsing and action interpretation; thus, commonly suffer from inference inefficiency and error propagation. Although there are studies that have considered multimodal architecture to process inputs in different modalities (Sun et al., 2022), however, those studies still rely on fine-grained environment parsing to ensure competitive performance. In contrast, this work is established upon first principles thinking, which directly reads the UI without additional environment parsing and provides the action (e.g., action type, gesture coordinate, and typed text) that can be executed without needing any extra APIs.

## 3 METHODOLOGY

In this section, we will first introduce the basic concepts for the UI control task and then describe the design of our proposed Auto-UI framework.

### 3.1 PROBLEM FORMALIZATION

Given a user instruction (also known as a *goal*), the agent needs to complete the task with multiple steps of interactions. The entire process is called an *episode*, which is composed of a series of *screens*. For each step in the episode, the agent will be provided with a screenshot, and the agent is required to predict the action until the task is complete. Detailed examples can be found in Appendix A.2.

### 3.2 FRAMEWORK OVERVIEW

Auto-UI is a multimodal agent that decides what action to take given the input screenshot and a user instruction. To empower the agent's decision making capability, we introduce a chain-of-action approach by leveraging a series of intermediate previous action histories and future action plans to predict actions.

The model architecture of Auto-UI is illustrated in Figure 2. On a high level, Auto-UI consists of three stages. First, we acquire encoded features from both vision and language inputs. Specifically, the vision input, i.e., a screenshot, is encoded by a frozen vision encoder. Meanwhile, the language input, consisting of the goal and a chain of previous action histories—each history contains a tuple {action type, touch point, lift point, and typed text}, is encoded by a language encoder. Second, the

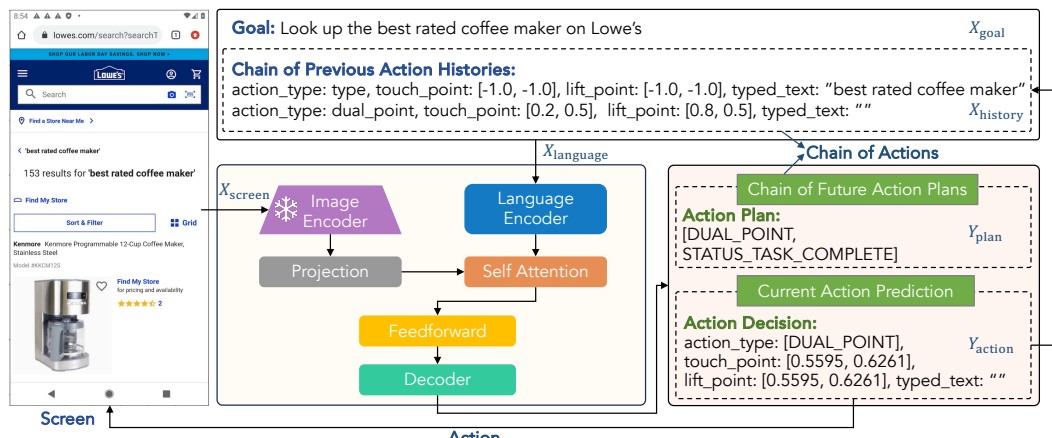

Figure 2: Model architecture of Auto-UI. A chain of action consists of a chain of previous action histories $X_{\text{history}}$ and a chain of future action plans $Y_{\text{plan}}$ in the illustration.

encoded vision and language representations are integrated by a self-attention module. Third, the fused representation is fed to the decoder to generate a chain of future action plans (i.e., action types to execute in future steps) followed by action prediction. A chain of action consists of two parts in the procedure above: a chain of previous action histories on the input side and a chain of future action plans on the output side. In the following, we describe the entire procedure in detail.

**Encoding** Suppose that an episode consists of $k$ steps of interactions. Given a screenshot $X_{\text{screen}} \in \mathbb{R}^{h \times w \times 3}$ with height $h$ and width $w$ at step $t \in [1, k]$, we first feed it to a frozen image encoder (e.g., BLIP-2 (Li et al., 2023)) and extract vision features $H_{\text{screen}} \in \mathbb{R}^{1 \times d_s}$ where $d_s$ is the dimension of the vision features. Additionally, we leverage a language encoder to extract the language features $H_{\text{language}} \in \mathbb{R}^{n \times d_l}$ of the input goal $X_{\text{goal}}$ where $n$ is the number of tokens and $d_l$ is the dimension of the language features. If $t > 1$, there will be a chain-of-action history already executed before step $t$. We denote the chain of action histories as $X_{\text{history}} = [m_1, \ldots, m_t]$ where $m_i$ contains a tuple of action type, touch point, lift point, and typed text. Otherwise, if $t = 1$, $X_{\text{history}}$ will be set empty:

$$X_{\text{history}} = \begin{cases} [m_1, \ldots, m_t], & \text{if } t > 1 \\ \text{<empty>}, & \text{otherwise} \end{cases} \tag{1}$$

We concatenate $X_{\text{goal}}$ and $X_{\text{history}}$ as the input to the language encoder: $X_{\text{language}} = \{X_{\text{goal}}, X_{\text{history}}\}$.

Then, we obtain the encoded representations of the vision and language inputs as follows:

$$H_{\text{screen}} = \text{VisionExtractor}(X_{\text{screen}}), \tag{2}$$

$$H_{\text{screen}}^{'} = W H_{\text{screen}}, \tag{3}$$

$$H_{\text{language}} = \text{LanguageEncoder}(X_{\text{language}}), \tag{4}$$

where $W$ is a trainable projection matrix to convert $H_{\text{screen}}$ into the same dimensionality as $H_{\text{language}}$.

**Interaction** We correlate $H_{\text{screen}}^{'}$ and $H_{\text{language}}$ with a single-head self-attention network (Vaswani et al., 2017), where the query ($Q$), key ($K$), and value ($V$) are $H_{\text{language}}$, $H_{\text{screen}}^{'}$, and $H_{\text{screen}}^{'}$, respectively. The attention output $H_{\text{screen}}^{\text{attn}} \in \mathbb{R}^{n \times d}$ is defined as: $H_{\text{screen}}^{\text{attn}} = \text{Softmax}(\frac{QK^{\top}}{\sqrt{d_k}})V$, where $d_k$ is the same as the dimension of $H_{\text{language}}$ because a single head is used.

Then, a gated fusion mechanism is adopted following prior studies (Zhang et al., 2020; Wu et al., 2021; Zhang et al., 2023b) to fuse $H_{\text{language}}$ and $H_{\text{screen}}^{\text{attn}}$. We have the fused output $H_{\text{fuse}} \in \mathbb{R}^{n \times d}$ by:

$$\lambda = \text{Sigmoid}(W_l H_{\text{language}} + W_v H_{\text{vision}}^{\text{attn}}), \tag{5}$$

$$H_{\text{fuse}} = (1 - \lambda) \cdot H_{\text{language}} + \lambda \cdot H_{\text{vision}}^{\text{attn}}, \tag{6}$$

where $W_l$ and $W_v$ are learnable parameters.

**Decoding** The fused representation $H_{\text{fuse}}$ is fed to a Transformer decoder to generate the target predictions in a string format. The target predictions consist of a chain of future action plans $Y_{\text{plan}}$ and the current action prediction $Y_{\text{action}}$ separated by specific prompts: {Action Plan: $Y_{\text{plan}}$, Action Decision: $Y_{\text{action}}$}. Concretely, $Y_{\text{plan}}$ is a chain of action types to execute in future steps: $Y_{\text{plan}}$ = [action_type$_t$, . . . , action_type$_k$]. $Y_{\text{action}}$ contains four components: $Y_{\text{action}}$ = {"action_type": <action_type>, "touch_point": <touch_point>, "lift_point": <lift_point>, "typed_text": <typed_text>}. These four components will be explained in the following subsection.

### 3.3 COORDINATE NORMALIZATION

Recall that a target action consists of four components: action type, touch point, lift point, and typed text. We consider six action types: *dual-point gesture*, *type*, *go_back*, *go_home*, *enter*, and *status_complete*. A dual-point gesture comprises a touch point and a lift point with $[y, x]$ coordinates. The gesture actions ensure a flexible action space and can represent clicks and scrolls at arbitrary locations. For example, a gesture action {"touch_point": [0.7761, 0.7089], "lift_point": [0.7761, 0.7089]} means clicking at the coordinate [0.7761, 0.7089], while a gesture action {"touch_point": [0.1898, 0.4477], "lift_point": [0.8242, 0.4077]} means scrolling down. A type action means typing a text and the text is placed in the <typed_text> field. The other action types, i.e., go_back, go_home, enter, and status_complete are system actions, whose corresponding <touch_point>, <lift_point> fields are filled with -1, and the <typed_text> is empty.

We observe that high-precision coordinates are not necessary for representing a click or scroll action. Therefore, we apply normalized values of the coordinates, which helps accelerate convergence and mitigate the ambiguity of coordinates. The normalization is applied to click and scroll actions. For click actions, we keep four decimal places. For scroll actions, we first determine the scroll direction with the touch point and lift point. Then, we transform the touch and lift points into fixed directional coordinates as follows: "up": {[0.8, 0.5], [0.2, 0.5]}, "down": {[0.2, 0.5], [0.8, 0.5]}, "left": {[0.5, 0.8], [0.5, 0.2]}, "right": {[0.5, 0.2], [0.5, 0.8]}, where {[·], [·]} consists of the touch point and lift point in the first [·] and second [·]. We provide examples of target actions in Appendix A.3.

## 4 EXPERIMENTS

### 4.1 DATASET

We use the AITW benchmark dataset (Rawles et al., 2023). AITW is a large-scale benchmark dataset for UI control, which contains natural language instructions, screenshots, and actions. There are $715K$ episodes spanning $30K$ unique instructions, covering diverse multi-step tasks such as application operation, web searching, and web shopping, on over 350 Apps and websites. This dataset covers various device types and operation systems in varying screen resolutions to ensure generality. There are five subsets in the benchmark dataset, namely, General, Install, GoogleApps, Single, and WebShopping. The details of the subsets and data statistics are presented in Appendix A.1.

### 4.2 BASELINES

We adopt three types of baselines for comparisons. The baselines encompass the In-context Learning (ICL) and fine-tuning paradigms, along with various backbone models of different sizes. This choice of baselines allows for a comprehensive comparison with our proposed approach.

(i) In-context Learning LLMs. Few-shot PaLM 2, ChatGPT (turbo-3.5) are adopted. Following previous studies (Rawles et al., 2023; Wang et al., 2023a), we feed the LLM a textual description of the screen and a user instruction. The textual description of the screen is formatted as an HTML syntax, providing the information of UI elements derived from OCR detection and icon detection from external tools (Rawles et al., 2023). The model is required to predict an action among pre-defined actions. If the action is clicking, the model will be required to provide the index of the clicked UI element. Alternatively, the model needs to provide the scroll direction if the action is scrolling. In addition, 5-shot CoT prompting is leveraged to improve the performance (Appendix A.4). In addition, we report the results of the multimodal GPT-4V by taking the vision image and action history as the input based on Yan et al. (2023).

(ii) Fine-tuned LLMs. We adopt Llama 2 (Touvron et al., 2023) as the baseline and fine-tune it with LoRA. We feed the model with the user instruction and the screen descriptions in HTML syntax (the same as adopted for in-context learning LLMs). The model is expected to predict the action in the same output format as in-context learning LLMs. As fine-tuning an LLM is expensive, we randomly sample 1% training data to help the LLM adapt to our tasks.

(iii) Specialized UI Agent. We adopted the Behavioural Cloning (BC) agent, which reported the state-of-the-art performance in Rawles et al. (2023). BC is a Transformer-based architecture that takes a task instruction, the current screen, and a stacked history of screen observations and actions as input. The task instruction and OCR-detected texts are encoded by a pre-trained BERT. The icons are represented by the embeddings for each of the bounding box points. The screen history is modeled by the $\{x, y\}$ positions of the touch and lift actions. All the embedded representations are fused to predict the action by a decoder. There are two BC variants, BC-single and BC-history, depending on whether the model takes as input the screen-action history.

## 4.3 Evaluation Measures

We compute the screen-wise action matching score as the main evaluation measure, defined as the number of correct actions divided by the episode length. A predicted action is considered correct if the action type and dual-point gesture match the gold ones. As we described in Section 3.3, the gesture actions can represent the click actions and scroll actions at arbitrary locations. Following Rawles et al. (2023), a click action is considered correct if its touch point and lift point fall within a 14% screen distance from the gold gestures or occur within the same detected bounding box with the gold gestures. A scroll action is considered correct if it has the same scroll axis as the gold gesture.

The screen-wise action matching score has been shown to correlate with the task complete score estimated by human evaluations (Rawles et al., 2023) and is appropriate to measure the action success rate for user instructions. Besides the overall matching score, we will also compare the click region accuracy, scroll direction accuracy, action type accuracy, and typed text accuracy for a more comprehensive reference (Section 5.1).

The evaluation criteria apply to the BC baselines and our Auto-UI. For the LLMs, they can only click on detected UI elements, rather than clicking at arbitrary locations. Therefore, we consider if the clicked UI element is matched for click actions instead of comparing dual-point gestures for LLMs.

## 4.4 Implementation Details

We adopt the encoder-decoder architecture (Raffel et al., 2020) under small (60M), base (200M) and large (700M) settings in our framework. We apply FLAN-Alpaca to initialize our model weights.[1] The vision features are obtained by the frozen BLIP-2 encoder (Li et al., 2023) (version: blip2_t5_instruct). We fine-tune the models up to 10 epochs, with a learning rate of 1e-4. The maximum input sequence length is 512. The batch size is 4. Our experiments are run on 8 NVIDIA Tesla V100 32G GPUs. Training the large and base models takes 75 and 25 hours, respectively.

We develop two kinds of approaches to analyze their generalization abilities, namely Auto-UI$_{\text{separate}}$, and Auto-UI$_{\text{unified}}$. Specifically, Auto-UI$_{\text{separate}}$ is trained and evaluated independently on each subset. Auto-UI$_{\text{unified}}$ is a unified model trained on the training sets of each subset and evaluated on each test set. As the GoogleApps subset is 10-100 times larger than the other subsets, using all the training data to train a unified model would suffer from the data imbalance issue (Zhang et al., 2022). Therefore, we only use 10% training data of GoogleApps. At the same time, the overall computation cost can also be saved by 80%. We use Auto-UI$_{\text{unified}}$ as the default model for analysis unless otherwise stated.

## 4.5 Main Results

Table 1 shows the main results. Auto-UI$_{\text{unified}}$ achieves the best overall performance compared with all the baselines. When compared with separate (not unified) models, Auto-UI$_{\text{unified}}$ shows general effectiveness across various task scenarios. The results show that a unified multimodal model out of *first principles thinking* can serve as a strong autonomous agent. Compared with previous BC models, Auto-UI$_{\text{unified}}$ has two major advantages. First, Auto-UI$_{\text{unified}}$ is a unified model that can be adapted

---

[1]https://github.com/declare-lab/flan-alpaca.

Table 1: Main results (%). Segment 1: specialized agent baselines; Segment 2: in-context learning LLM baselines; Segment 3: fine-tuned Llama 2 baseline; Segment 4: our Auto-UI results. Prior published best results are marked with an underline. "Unified" means a general model that can work across subsets. "w/o Anno." means no screen description is needed. The PaLM-CoT and BC results are from Rawles et al. (2023). The GPT-4V result is from Yan et al. (2023). The other results are based on our own implementations. The overall score is computed as the average accuracy on all the subsets. The best average result is in **bold** face.

| Model | Unified | w/o Anno. | Overall | General | Install | GoogleApps | Single | WebShopping |
|---|---|---|---|---|---|---|---|---|
| PaLM 2-CoT | ✓ | ✗ | 39.6 | - | - | - | - | |
| ChatGPT-CoT | ✓ | ✗ | 7.72 | 5.93 | 4.38 | 10.47 | 9.39 | 8.42 |
| GPT-4V | ✓ | ✗ | 52.96 | 43.01 | 46.14 | 49.18 | 78.29 | 48.18 |
| Fine-tuned Llama 2 | ✗ | ✗ | 28.40 | 28.56 | 35.18 | 30.99 | 27.35 | 19.92 |
| BC-single | ✗ | ✗ | 68.7 | - | - | - | - | |
| BC-history | ✗ | ✗ | 73.1 | 63.7 | 77.5 | 75.7 | 80.3 | 68.5 |
| Auto-UI$_{separate}$ | ✗ | ✓ | 74.07 | 65.94 | **77.62** | **76.45** | 81.39 | 69.72 |
| Auto-UI$_{unified}$ | ✓ | ✓ | **74.27** | **68.24** | 76.89 | 71.37 | **84.58** | **70.26** |

Table 2: Ablation study of Auto-UI design components. We adopt Auto-UI$_{unified}$ for analysis.

| Model | Overall | General | Install | GoogleApps | Single | WebShopping |
|---|---|---|---|---|---|---|
| Auto-UI | **74.27** | **68.24** | **76.89** | **71.37** | **84.58** | **70.26** |
| w/o chain of actions | 68.53 | 58.99 | 72.06 | 67.50 | 81.25 | 62.86 |
| w/ previous action history | 73.78 | 67.97 | 76.66 | 71.00 | 83.64 | 69.62 |
| w/ future action plan | 68.81 | 59.01 | 72.34 | 67.95 | 81.53 | 63.24 |
| w/o coordinate normalization | 70.23 | 63.79 | 73.28 | 66.63 | 82.11 | 65.33 |

to different scenarios without the need to train specific models for each task. Second, Auto-UI$_{unified}$ does not need additional annotations (screen parsing) and is easy to use. We will provide a more detailed analysis of the generality of computation efficiency in Section 5.2 and 5.4.

The ablation study in Table 2 verifies that both the chain of actions and coordinate normalization contribute to the overall performance (+5.74% and 4.04%, respectively). We set the maximum numbers of the previous actions and future actions to 8 and 4, respectively. The choice is made according to our analysis on the General subset with Auto-UI$_{separate}$ (Figure 3). The model under those setups achieves the optimal performance and both the input and output sequence lengths would not exceed the model limit.

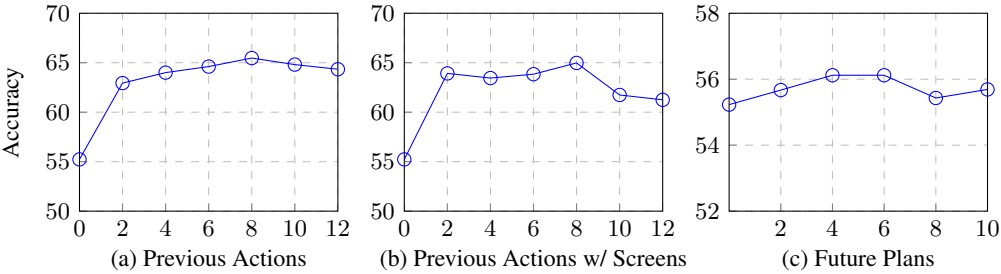

Figure 3: Performance of Auto-UI with respect to varying numbers of chains of actions.

For the LLMs, using either prompting or fine-tuning techniques does not achieve competitive performance compared with the other approaches. The most plausible reason is that they learn from the parsed HTML elements of the screen so that they may suffer from information loss compared with more informative vision features of the screens. Specifically, we find that ChatGPT is quite accurate at predicting the action type but fails at lower-level executions (Appendix B.1).

It is reasonable that Auto-UI$_{\text{unified}}$ performs relatively inferior to BC-history on the two App-centered subsets, Install and GoogleApps, because we only use 10% training data of GoogleApps considering the data balance and computation overhead. We observe that the performance does not improve when we use all the training data of GoogleApps, possibly due to the data imbalance issue (Zhang et al., 2022). In contrast, our separate model Auto-UI$_{\text{separate}}$ can achieve better performance than BC-history, showing that our approach is better than BC-history under the same training setting. As we aim to study a simple and unified approach that achieves generally strong performance, we leave the treatment of the data imbalance issue in future work.

# 5    ANALYSIS

## 5.1    CATEGORY ACCURACY

To dive into the capability of Auto-UI, we calculate the click region accuracy, scroll direction accuracy, action type accuracy, and typed text accuracy. Figure 4 presents the results. We see that Auto-UI achieves over 90% action type accuracy on average. In contrast, the major challenges lie within the click region and scroll direction predictions. Although the model is able to predict the right action most of the time, it tends to click a wrong place or scroll in a wrong direction. The result reveals a future direction of improving the model's ability to understand the screen layouts, e.g., using more advanced vision features.

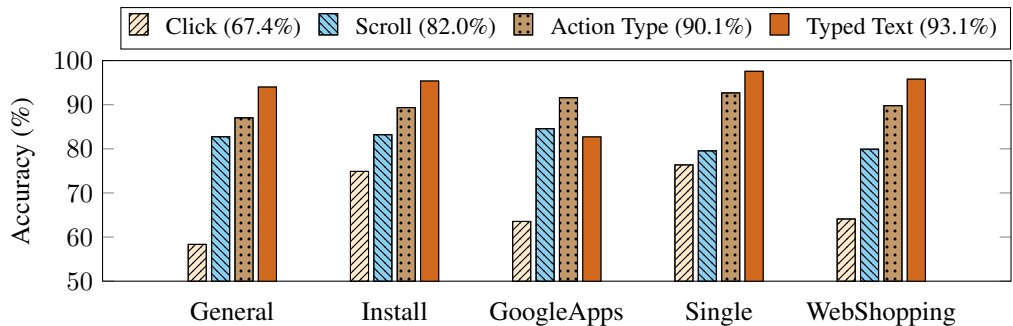

Figure 4: Category accuracy of our Auto-UI. The values in parentheses represent the average category accuracy on the subsets.

## 5.2    GENERALIZATION ABILITY

As our approach is designed under first principles thinking and does not rely on pre-defined internal APIs, it could be easily generalized to new task domains. To verify the generality, we evaluate the performance of Auto-UI$_{\text{separate}}$ on each subset in Figure 5. For example, we train an Auto-UI$_{\text{separate}}$ model on the training set of General and then test its performance on the tests of each subset. We see that our approach is able to achieve a decent performance though the domains vary. This result reveals that the model could capture general knowledge for the UI control task; thus is applicable to different domains. In addition, the unified model Auto-UI$_{\text{unified}}$ can serve as a potential choice in real-world applications owing to more coverage of training data.

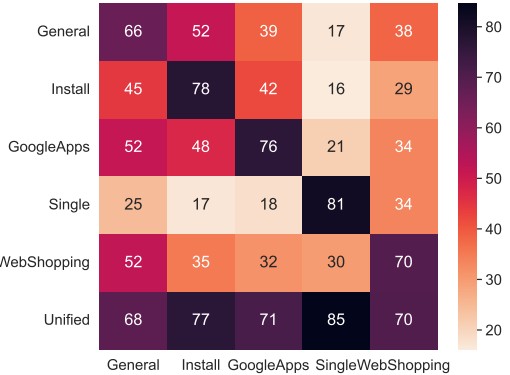

Figure 5: Dataset transfer results of Auto-UI.

## 5.3 Comprehensive Analysis

Here we present a comprehensive analysis of the choice of pre-trained features and model scale. The results are summarized in Table 3.

Table 3: Results varying vision features and pre-trained language model weights.

| Model | Overall | General | Install | GoogleApps | Single | WebShopping |
|---|---|---|---|---|---|---|
| Auto-UI on CLIP | 71.84 | 66.28 | 74.40 | 69.71 | 81.60 | 67.23 |
| Auto-UI on BLIP-2 | 74.27 | 68.24 | 76.89 | 71.37 | 84.58 | 70.26 |
| Auto-UI on Vanilla-T5$_{large}$ | 72.98 | 66.61 | 75.40 | 70.86 | 83.47 | 68.54 |
| Auto-UI on FLAN-T5$_{large}$ | 73.36 | 67.59 | 76.35 | 70.71 | 83.01 | 69.12 |
| Auto-UI on FLAN-Alpaca$_{large}$ | 74.27 | 68.24 | 76.89 | 71.37 | 84.58 | 70.26 |
| Auto-UI on FLAN-Alpaca$_{small}$ | 71.38 | 65.26 | 74.90 | 68.70 | 81.20 | 66.83 |
| Auto-UI on FLAN-Alpaca$_{base}$ | 72.84 | 66.97 | 75.93 | 70.29 | 82.56 | 68.46 |
| Auto-UI on FLAN-Alpaca$_{large}$ | 74.27 | 68.24 | 76.89 | 71.37 | 84.58 | 70.26 |

• Pre-trained Features. There are two kinds of pre-trained features used in this work, the vision features and language model weights. For vision features, we compare two popular types, CLIP (Radford et al., 2021) and BLIP-2 (Li et al., 2023). We observe that BLIP-2 achieves relatively better performance. Therefore, we use BLIP-2 by default in Auto-UI. For pre-trained language model weights, we compare initializing the model with the vanilla T5 (Raffel et al., 2020), FLAN-T5 (Chung et al., 2022), and FLAN-Alpaca (Taori et al., 2023a) weights under the large size. We see that FLAN-Alpaca achieves the best performance as it has been optimized with Stanford Alpaca synthetic instruction tuning data.

• Model Scale. Compared with the performance gains from our technique components (chain of actions and coordinate normalization) in Table 2, the benefit of scaling parameter size becomes relatively marginal. As we observe that a larger model size does not lead to dramatic improvement in performance, we do not scale the model scale but focus on the base (220M) and large (770M) models in this work. In addition, our choice is also based on other considerations, including the constriction of GPU memory and computation budget.

## 5.4 Computation Cost

Table 4 compares the inference speed and GPU memory cost for Auto-UI and Llama 2. Auto-UI is able to achieve nearly real-time inference (within less than one second for an action prediction) with less than 10GB GPU memory. The inference speed is over 10 times faster than Llama 2. Our work shows the strength of the medium-sized language model in building autonomous agents, which is able to achieve competitive performance with fast inference.

Table 4: Computations cost of Auto-UI and Llama. The computation efficiency is computed by time (s) divided by the number of inferences (n). Llama 2 is hosted with 8-bit quantization and float16 precision to improve the inference speed.

| Model | Feature Extraction (s/n) | Model Inference (s/n) | Peak GPU Memory (GB) |
|---|---|---|---|
| Auto-UI$_{base}$ | 0.06 | 0.19 (45x) | 4.6 (10x) |
| Auto-UI$_{large}$ | 0.06 | 0.59 (15x) | 8.2 (6x) |
| Llama 2 | - | 8.5 | 49.7 |

## 6 Conclusion

This work presents an autonomous UI agent called Auto-UI that can interact in a multimodal UI environment without environment parsing or application-dependent API access. In addition, we propose a chain-of-action technique that leverages the previously executed actions and future action plans to help the agent decide what action to execute. Experimental results show that Auto-UI achieves superior performance to previous prompting-based and fine-tuning baselines. Besides the strong performance and generality across domains, Auto-UI can infer an action as fast as within less than one second.

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

## A APPENDIX

### A.1 DATA STATISTICS

We use the AITW benchmark dataset (Rawles et al., 2023). AITW is a large-scale benchmark dataset for UI control, which contains natural language instructions, screenshots, and actions. There are $715K$ episodes spanning $30K$ unique instructions, covering diverse multi-step tasks such as application operation, web searching, and web shopping, on over 350 Apps and websites. This dataset covers various device types and operation systems in varying screen resolutions to ensure generality. There are five subsets in the benchmark dataset, namely, General, Install, GoogleApps, Single, and WebShopping.

(i) General contains miscellaneous tasks that need interaction with third-party Apps and websites, as well as question answering.

(ii) Install contains tasks related to installing and uninstalling Apps, App login, and App login support.

(iii) GoogleApps contains tasks about manipulating various Google applications such as Gmail, Calendar, Photos, and Settings.

(iv) Single contains atomic tasks (e.g., "upvote the post") whose preceding actions have been already completed (e.g., opening Instagram, going to home feed, looking at a post).

(v) WebShopping contains tasks related to online shopping on E-commerce websites, e.g., searching for an item, adding an item to the cart, and viewing the shopping cart.

Table 5 presents the data statistics of the AITW dataset. Each subset is split episode-wise into a training, validation, and test set (80/10/10%).

Table 5: Dataset statistics.

| Dataset | Episodes | Screens | Instructions |
|---|---|---|---|
| General | 9,476 | 85,413 | 545 |
| Install | 25,760 | 250,058 | 688 |
| GoogleApps | 625,542 | 4,903,601 | 306 |
| Single | 26,303 | 85,668 | 15,366 |
| WebShopping | 28,061 | 365,253 | 13,473 |

### A.2 TASK EXAMPLES

We show the task examples from the AITW benchmark dataset (Rawles et al., 2023). Figures 6-10 show the examples in each subset, i.e., General, Install, GoogleApps, Single, and WebShopping. The gold actions for each screen are depicted in the illustrations for reference.

Goal: Open a new Chrome private window

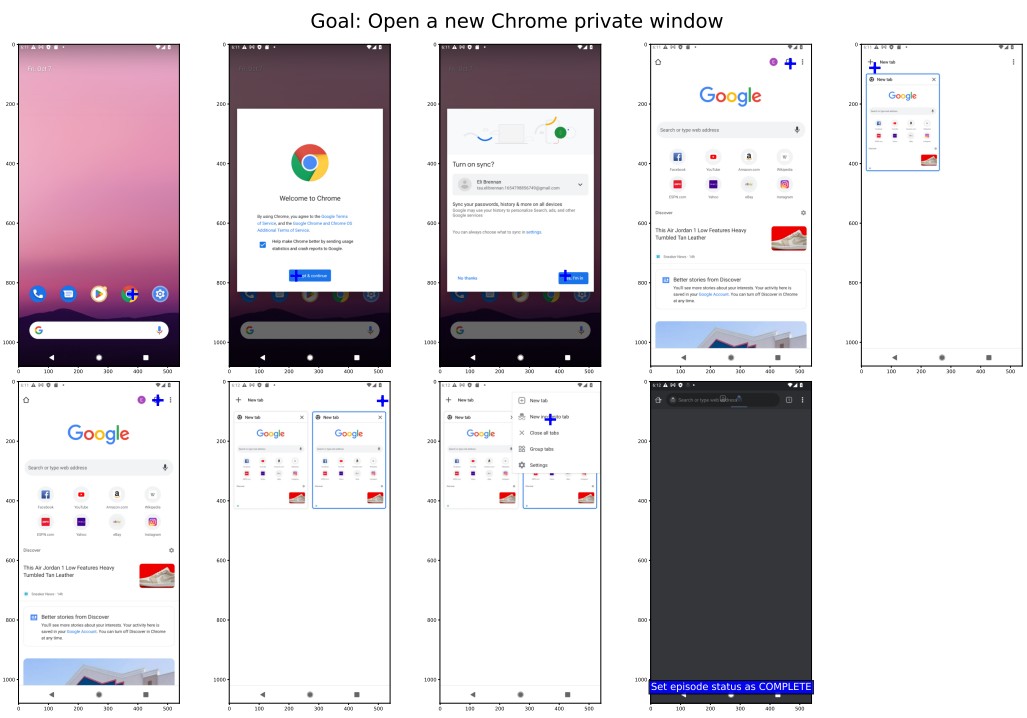

Figure 6: An example episode from General.

Goal: uninstall "Microsoft Authenticator"

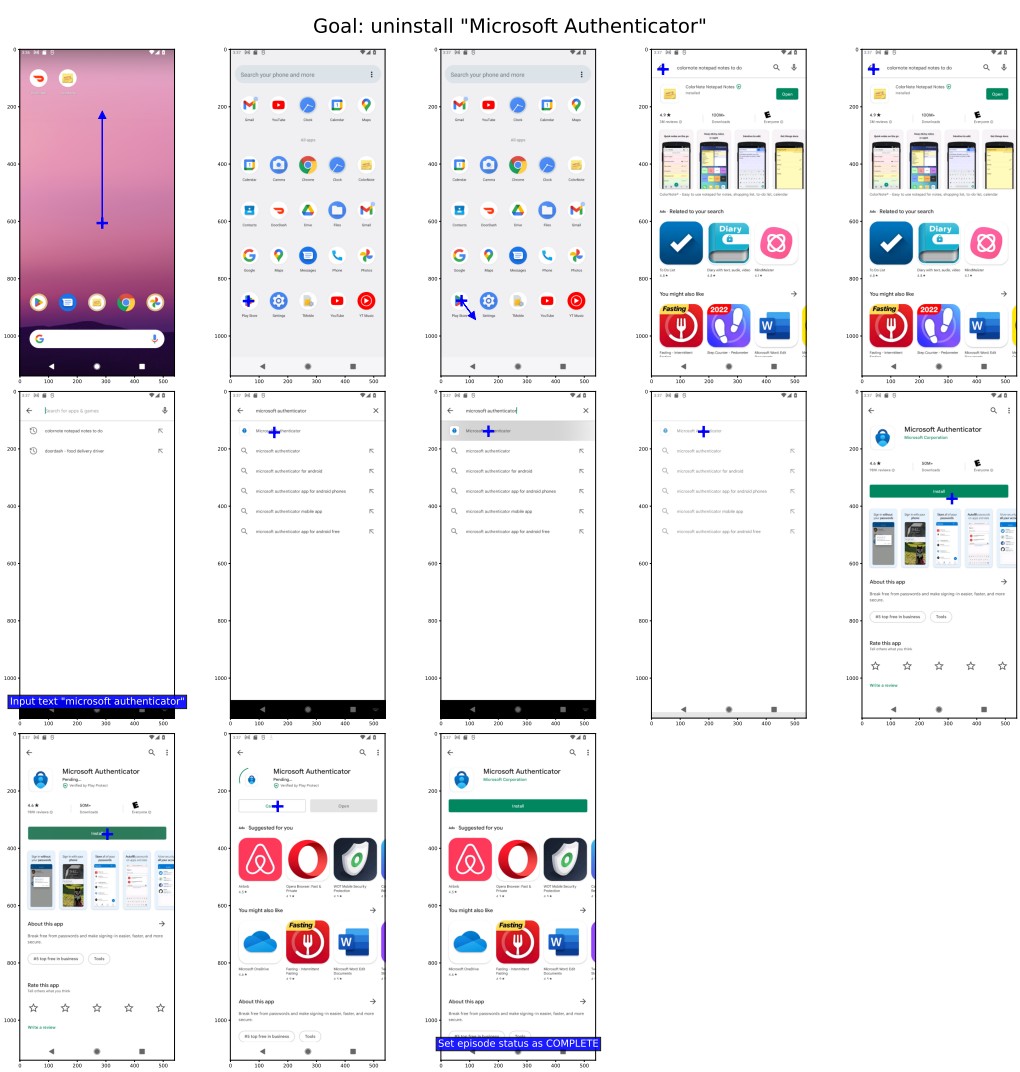

Figure 7: An example episode from Install.

Goal: turn off javascript in the chrome app

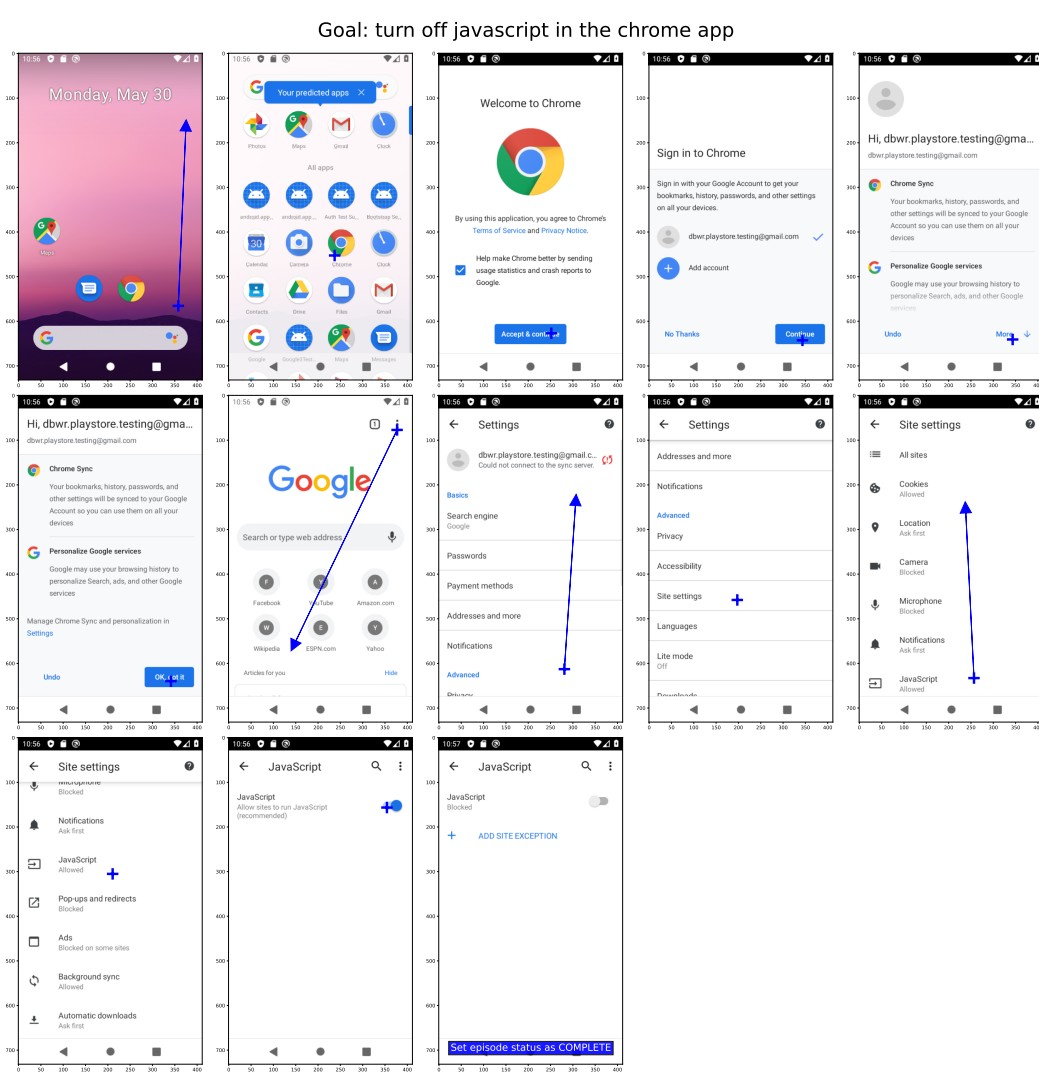

Figure 8: An example episode from GoogleApps.

Goal: go to google search bar and search & open ebay.com in chrome

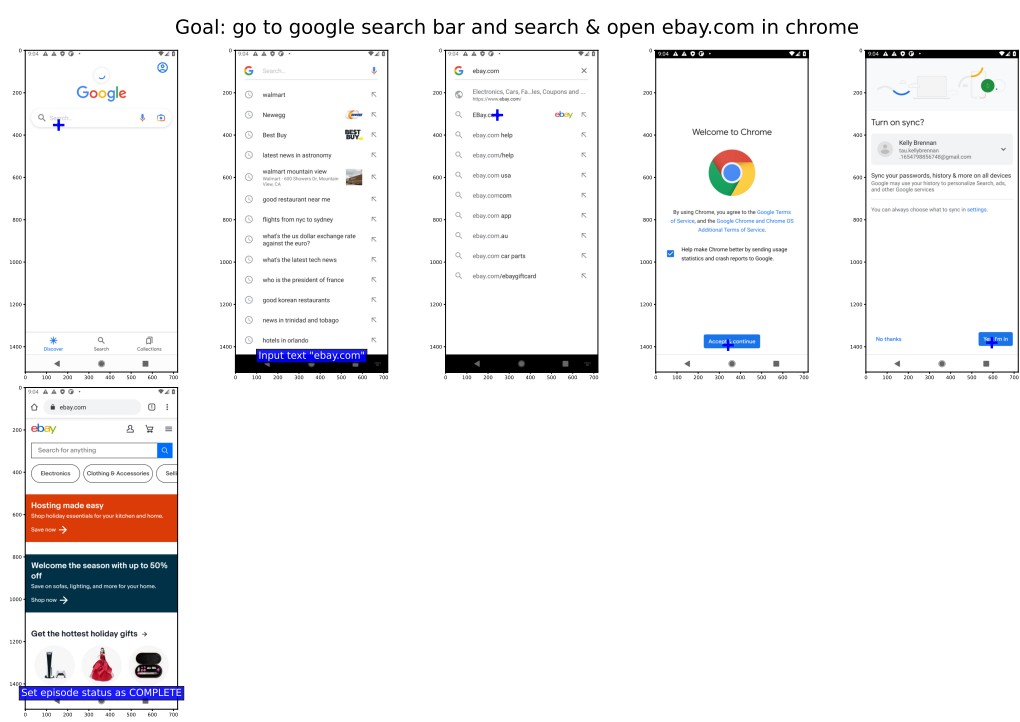

Figure 9: An example episode from Single.

Goal: Look up the best rated coffee maker on Lowe's.

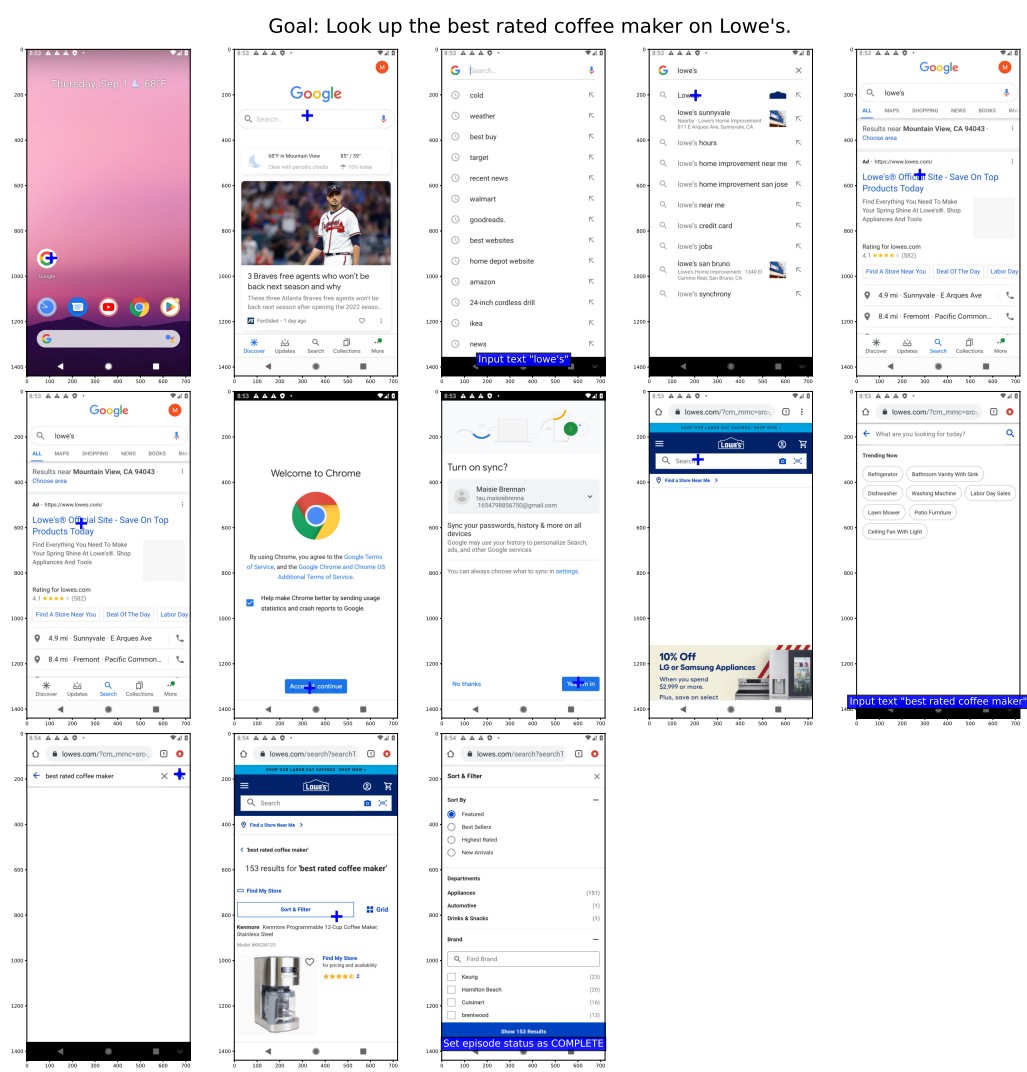

Figure 10: An example episode from WebShopping.

## A.3    COORDINATE NORMALIZATION

Table 6: Target output examples after the coordinate normalization.

| Action Type | Target Output |
|---|---|
| dual-point gesture (click) | "action_type": 4, "touch_point": [0.8497, 0.5964], "lift_point": [0.8497, 0.5964], "typed_text": "" |
| dual-point gesture (scroll) | "action_type": 4, "touch_point": [0.2, 0.5], "lift_point": [0.8, 0.5], "typed_text": "" |
| type | "action_type": 3, "touch_point": [-1.0, -1.0], "lift_point": [-1.0, -1.0], "typed_text": "what's the news in chile?" |
| go_back | "action_type": 5, "touch_point": [-1.0, -1.0], "lift_point": [-1.0, -1.0], "typed_text": "" |
| go_home | "action_type": 6, "touch_point": [-1.0, -1.0], "lift_point": [-1.0, -1.0], "typed_text": "" |
| enter | "action_type": 7, "touch_point": [-1.0, -1.0], "lift_point": [-1.0, -1.0], "typed_text": "" |
| status_complete | "action_type": 10, "touch_point": [-1.0, -1.0], "lift_point": [-1.0, -1.0], "typed_text": "" |

## A.4    LLM PROMPT

We use the following prompt for PaLM 2-CoT and ChatGPT-CoT due to its optimal performance reported in Rawles et al. (2023).

```
Given a mobile screen and a question, provide the action based on the screen information.

Available Actions:
{"action_type": "click", "idx": <element_idx>}
{"action_type": "type", "text": <text>}
{"action_type": "navigate_home"}
{"action_type": "navigate_back"}
{"action_type": "scroll", "direction": "up"}
{"action_type": "scroll", "direction": "down"}
{"action_type": "scroll", "direction": "left"}
{"action_type": "scroll", "direction": "right"}

Previous Actions:
{"step_idx": 0, "action_description": "press [HOME key]"}
{"step_idx": 2, "action_description": "click [Google Icon]"}
{"step_idx": 3, "action_description": "click [search for hotels]"}

Screen:
 </img>
 </img>
<p id=2 class="text" alt="search for hotels"> search for hotels </p>
<p id=3 class="text" alt="in"> in </p>
<p id=4 class="text" alt="mexico city mexico"> mexico city mexico </p>
 </img>
<p id=6 class="text" alt="Share"> Share </p>
<p id=7 class="text" alt="Select alI"> Select alI </p>
<p id=8 class="text" alt="Cut"> Cut </p>
<p id=9 class="text" alt="Copy"> Copy </p>
<p id=10 class="text" alt="hotel in mex"> hotel in mex </p>
 </img>
<p id=12 class="text" alt="best hotel"> best hotel </p>
<p id=13 class="text" alt="mexico city"> mexico city </p>
<p id=14 class="text" alt="in"> in </p>
 </img>
<p id=16 class="text" alt="K"> K </p>
<p id=17 class="text" alt="hotel ciudad"> hotel ciudad </p>
<p id=18 class="text" alt="de mexico"> de mexico </p>
<p id=19 class="text" alt="gran"> gran </p>
 </img>
 </img>
 </img>

Instruction: What time is it in Berlin?
Answer: Let's think step by step. I see unrelated search results in the Google app,
I must clear the search bar, so the action is {"action_type": "click", "idx": 1}

Previous Actions:
```

```
{"step_idx": 0, "action_description": "click [DISMISS]"}

Screen:
<p id=0 class="text" alt="Update your"> Update your </p>
<p id=1 class="text" alt="Gmail app"> Gmail app </p>
<p id=2 class="text" alt="attach files from"> attach files from </p>
<p id=3 class="text" alt="To"> To </p>
<p id=4 class="text" alt="download the"> download the </p>
<p id=5 class="text" alt="Drive,"> Drive, </p>
<p id=6 class="text" alt="latest"> latest </p>
<p id=7 class="text" alt="version"> version </p>
<p id=8 class="text" alt="of"> of </p>
<p id=9 class="text" alt="Gmail"> Gmail </p>
<p id=10 class="text" alt="UPDATE"> UPDATE </p>
<p id=11 class="text" alt="DISMISS"> DISMISS </p>
<p id=12 class="text" alt="Got"> Got </p>
<p id=13 class="text" alt="it"> it </p>
 </img>

Instruction: see creations saved in the google photos
Answer: Let's think step by step. I see a popup, I need to open Google Photos, so
the action is {"action_type": "click", "idx": 11}

Previous Actions:

Screen:
<p id=0 class="text" alt="M"> M </p>
<p id=1 class="text" alt="New in Gmail"> New in Gmail </p>
<p id=2 class="text" alt="All the features you"> All the features you </p>
<p id=3 class="text" alt="love with"> love with </p>
<p id=4 class="text" alt="a fresh"> a fresh </p>
<p id=5 class="text" alt="look"> look </p>
<p id=6 class="text" alt="new"> new </p>
<p id=7 class="text" alt="GOT IT"> GOT IT </p>

Instruction: open app "Google Play services"
Answer: Let's think step by step. I see the GMail app, I need to open the app
drawer, so the action is {"action_type": "navigate_home"}

Previous Actions:

Screen:
<p id=0 class="text" alt="Tuesday, Aug"> Tuesday, Aug </p>
<p id=1 class="text" alt="9"> 9 </p>
 </img>
 </img>

Instruction: open app "Messenger Lite" (install if not already installed)
Answer: Let's think step by step. I see the home screen, I need to open the app
drawer, I should swipe up, so the action is {"action_type": "scroll", "direction":
"down"}

Previous Actions:
{"step_idx": 0, "action_description": "scroll down"}

Screen:
 </img>
<p id=1 class="text" alt="Search your phone and more"> Search your phone and more </p>
<p id=2 class="text" alt="M"> M </p>
<p id=3 class="text" alt="O"> O </p>
 </img>
<p id=5 class="text" alt="Clock"> Clock </p>
<p id=6 class="text" alt="YouTube"> YouTube </p>
<p id=7 class="text" alt="Photos"> Photos </p>
<p id=8 class="text" alt="Gmail"> Gmail </p>
<p id=9 class="text" alt="All apps"> All apps </p>
<p id=10 class="text" alt="g"> g </p>
<p id=11 class="text" alt="O"> O </p>
 </img>
<p id=13 class="text" alt="10"> 10 </p>
<p id=14 class="text" alt="Calendar"> Calendar </p>
<p id=15 class="text" alt="Camera"> Camera </p>
<p id=16 class="text" alt="Chrome"> Chrome </p>
<p id=17 class="text" alt="Clock"> Clock </p>
<p id=18 class="text" alt="O"> O </p>
<p id=19 class="text" alt="M"> M </p>
<p id=20 class="text" alt="B"> B </p>
 </img>
<p id=22 class="text" alt="Gmail"> Gmail </p>
<p id=23 class="text" alt="Drive"> Drive </p>
<p id=24 class="text" alt="Files"> Files </p>
```

```
<p id=25 class="text" alt="Contacts"> Contacts </p>
<p id=26 class="text" alt="G OO"> G OO </p>
 </img>
 </img>
 </img>
 </img>
<p id=31 class="text" alt="Google"> Google </p>
<p id=32 class="text" alt="Maps"> Maps </p>

Instruction: Search for hotels in Chicago.
Answer: Let's think step by step. I see the app drawer, I need to search, so the
action is {"action_type": "click", "idx": 27}

Previous Actions:
<HISTORY>
Screen:
<SCREEN_REPRESENTATION>
Instruction: <GROUNDING_GOAL>
Answer: Let's think step by step. I see
```

## A.5 USING SCREEN DESCRIPTIONS

We are interested in whether Auto-UI can be further improved when screen annotations are available. Therefore, we incorporate screen descriptions containing icon and text information, organized in HTML syntax, into our language input $X_{\text{language}}$. Detailed examples of screen descriptions can be found in the "Screen" section in A.4.

Table 7: Results of Auto-UI when using annotated screen descriptions.

| Model | Overall | General | Install | GoogleApps | Single | WebShopping |
|---|---|---|---|---|---|---|
| Auto-UI$_{\text{base}}$ | 72.84 | 66.97 | 75.93 | 70.29 | 82.56 | 68.46 |
| w/ Screen Descriptions | 75.54 | 70.30 | 78.05 | 73.04 | 85.31 | 71.00 |

In Table 7, we see that Auto-UI can perform better when the annotated screen descriptions are available. The results show that there is still room for performance gains for Auto-UI. However, as the annotations are not always available in real-world applications, we do not include them by default in our framework.

## B FURTHER ANALYSIS

### B.1 CATEGORY COMPARISON WITH THE ICL BASELINE

To understand how the ICL baseline performs on our task and assess the advantage of Auto-UI, we conduct a category comparison with ChatGPT.

Table 8: Category comparison with the ICL baseline on the General test.

| Model | Overall | Action Type | Click | Scroll |
|---|---|---|---|---|
| ChatGPT | 5.93 | 41.72 | 8.50 | 4.00 |
| Auto-UI | 68.24 | 87.03 | 58.34 | 82.74 |

We see that the ICL method (ChatGPT) is quite accurate at predicting the action type (41.72%) but fails at lower-level executions, e.g., clicking positions (8.5%) and scrolling directions (4.0%). The results show that using HTML-based layout information is not enough to accurately execute actions. In contrast, Auto-UI has the advantage of predicting both action types and performing low-level executions by leveraging multimodal perception and the chain-of-action technique.

