# OpenReview forum: "You Only Look at Screens: Multimodal Chain-of-Action Agents"
_ICLR.cc/2024/Conference — Submitted to ICLR 2024_

### Official Review · Reviewer_BNVX · 2023-10-25

**Soundness:** 3 good
**Presentation:** 3 good
**Contribution:** 3 good
**Rating:** 6
**Confidence:** 3

**Summary:**

The paper proposes Auto-UI, a multimodal solution that directly interacts with the interface, which eliminates the need for environment parsing or reliance on application-specific APIs. The authors introduce a chain-of-action technique that incorporates previous action histories and future action plans to guide the agent's decision-making process. The approach is evaluated using a device-control benchmark called AITW, which consists of 30,000 unique instructions covering tasks like application operation, web searching, and web shopping. Experimental results demonstrate that Auto-UI achieves state-of-the-art performance, with high accuracy in predicting action types (90%) and an overall success rate of 74%. The authors have made the code available for review.

**Strengths:**

1. The proposed Auto-UI approach demonstrates a level of originality in addressing the challenges of autonomous user interface agents. By directly interacting with the interface instead of relying on environment parsing or application-specific APIs, it offers a novel solution that bypasses common inefficiencies and risks associated with existing approaches. The introduction of the chain-of-action technique also adds a unique element to the decision-making process of the agent.
2. The approach is evaluated through experiments with the AITW benchmark. The inclusion of 30,000 unique instructions covering various multi-step tasks provides a comprehensive assessment of the Auto-UI system. Achieving a state-of-the-art performance demonstrates the effectiveness and reliability of the proposed solution.
3. Overall, the paper is clear and easy to follow. The text provides a clear description of the challenges faced by existing approaches, introduces the Auto-UI solution, and explains the chain-of-action technique. The inclusion of experimental results contribute to a clear understanding of the proposed methodology and its performance.
4. By addressing the challenges of inference inefficiency and error propagation, Auto-UI offers a more efficient and reliable approach to task automation. The multimodal solution and the elimination of environment parsing and reliance on application-specific APIs provide a significant advancement in the development of autonomous UI agents. Furthermore, the state-of-the-art performance achieved on the AITW benchmark showcases the practical applicability and potential impact of the proposed approach.

**Weaknesses:**

1. While the authors highlight the chain-of-action technique as a contribution, it appears to primarily concatenate the output actions, which can be confusing. It would be helpful to provide a more detailed explanation or clarification of how the chain-of-action technique enhances the decision-making process and contributes to the overall effectiveness of the Auto-UI approach.

2. The experiment section lacks an explanation for the rationale behind selecting specific baselines. It would be valuable to include a justification for choosing the particular baselines used in the evaluation. Additionally, providing information on the performance of a GPT4 model, if available, would offer a useful benchmark to compare the performance of the proposed Auto-UI approach.

**Questions:**

GPT4 is reported to possess significantly improved agent capabilities compared to existing LLMs. However, it is important to note that the specific performance metrics and details of GPT4 have not been provided in the given context. Therefore, the performance of GPT4 remains unclear and unavailable for direct comparison in this discussion. What is the performance of GPT4?

---

> ### Author Response · Authors · 2023-11-19
>
> Thanks for your insightful review and constructive feedback.
>
> > W1: "While the authors highlight the chain-of-action technique as a contribution, it appears to primarily concatenate the output actions, which can be confusing. It would be helpful to provide a more detailed explanation or clarification of how the chain-of-action technique enhances the decision-making process and contributes to the overall effectiveness of the Auto-UI approach."
>
> Chain-of-action (CoA) technique contributes two aspects instead of purely concatenating the output actions. It organizes a chain of previous action histories on the input side and a chain of future action plans on the output side, which encourages the agent to leverage action history and make future plans before predicting the current action. The working mechanism can be seen as bidirectional.
>
> CoA helps improve the decision-making process and contributes to the overall effectiveness by +5.74% accuracy. To investigate the effectiveness, we conduct additional analysis to ablate the CoA technique on category accuracy.
>
> | Model                     | Overall | Action Type  | Click        | Scroll       | Text |
> | ------------------------- | ------- | ------- | ------- | ------- | ------- |
> | w/o Chain of Actions | 58.99 | 81.69 | 48.38 | 75.83 | 93.57 |
> | w/ Chain of Actions | 68.24  (+9.25) | 87.03  (+5.34) | 58.34 (+9.96) | 82.74  (+6.91)  | 93.99 (+0.42) |
>
> The results show that CoA helps improve the clicking and scrolling accuracy by a large margin (9.96 and 6.91), e.g., clicking accurate positions and scrolling in the correct direction, compared with the action type accuracy and text accuracy. The most possible reason is that CoA provides action history and future plans as the context for performing more accurate executions given that the model has already been effective in predicting the action type (over 81.69%).
>
> > W2 & Q: "The experiment section lacks an explanation for the rationale behind selecting specific baselines. It would be valuable to include a justification for choosing the particular baselines used in the evaluation. Additionally, providing information on the performance of a GPT4 model, if available, would offer a useful benchmark to compare the performance of the proposed Auto-UI approach."
>
> Yes. We have added a justification for the choice of the baselines in Section 4.2. The specialized UI agent is selected because it is the previous state-of-the-art approach in existing studies. The in-context learning LLMs are selected because they reflect the widely used paradigm of developing LLM agents. Fine-tuned LLMs are adopted to investigate the potential of leveraging open-source LLMs for our problem. The baselines encompass both the In-context Learning and fine-tuning paradigms, along with various backbone models of different sizes. These baselines were carefully chosen to ensure a thorough evaluation of our work's performance against existing approaches.
>
> Following the reviewer’s comment, we have added the performance of the GPT-4V model. The main results are shown below.
>
> | Model                     | Overall | General  |  Install  | GoogleApps  | Single  |
> | ------------------------- | ------- | ------- | ------- | ------- | ------- |
> | PaLM 2-CoT|  39.6 | -  |  - |  - |   -  |
> | ChatGPT  | 7.72  | 5.93  | 4.38  | 10.47  |  9.39  |  8.42  |
> | GPT-4V  | 52.96   | 43.01   | 46.14   | 49.18      | 78.29  |
> | Auto-UI | 74.27   | 68.24   | 76.89   | 71.37      | 84.58  |
>
> The results show that GPT-4V achieves much better performance than the PaLM and ChatGPT models. Auto-UI is still better than GPT-4V.

---

### Official Review · Reviewer_CMSv · 2023-10-26

**Soundness:** 3 good
**Presentation:** 3 good
**Contribution:** 3 good
**Rating:** 5
**Confidence:** 4

**Summary:**

This work proposes a “chain-of-action” approach to tackle the autonomous web-searching agent problem. Specifically, they propose a multimodal framework that firstly encodes both the language goals and the web-interaction histories, as well as the screen images, into a combined representation, where a decoder will generate a look-ahead future action plan and a formatted immediate next action to perform.
The authors conducted experiments on the AITW dataset where an AI agent is tasked to interact with a Web UI following certain goals, where they demonstrate the effectiveness of the proposed models against three major baselines.

**Strengths:**

- The proposed framework is claimed to be much lighter weight than methods that try to take the whole web information into textualized format for agents to comprehend.
- The formatted action is sound and should be generalizable to other web-search domains.
- The paper is pretty easy to follow, with illustrations onto the points.
- The generalization ablation studies are helpful to gauge the capacity of the proposed framework.

**Weaknesses:**

- The paper does not describe much about the actual training details, in that sense, to me, the proposed method is still a kind of BC, where the target decoding is optimized towards mimicking the golden action sequences. (Unless some RL or other mechanism is used here, which is not described.) In my opinion, the novelties here mainly lie in the multimodal representations (both modality taken into account) and the format of the action performed.
- I’m a bit skeptical about the ICL baseline, first of all more details (e.g., how actions are represented, how OCRed results are used) of that baseline need to be described, at least in the appendix. Secondly, it also needs to be evaluated at the action plan level, my guess is that this method should be quite accurate on those but might fail more on the lower-level executions. Thirdly, it is indeed unfair simply because the model is not taking the images into account, which could be the key towards the success of the proposed method in this work. So, at least a multimodal version of it needs to be taken into consideration, or, a better spatial representation of the html syntax is required. (HTML can be many times too coarse to represent a spatial layout.)
- Similar to above, the third baseline, fine-tuning LLMs, need to have a version with multimodal inputs.
- An error analysis is required both on the quantitative and qualitative sides, what are the major errors that these models exhibit?

**Questions:**

- I’m a bit surprised that the language decoder is able to predict tokens as precisely as four decimal places, or is the actual precision here not important? I.e., could you not simply split image screens into patches and just use their centers as the coordinate representations? (And the more patches you grid the screen, the more precise it would be.)
- What are the main types of errors observed by the proposed framework? And, does the framework provide good insights on how to assign these errors to specific modules? I.e., where should the improvements be?

**Details Of Ethics Concerns:**

None.

---

> ### Author Response · Authors · 2023-11-19
>
> Thanks for your insightful review and constructive feedback.
>
> > W1: Training details & Novelty.
>
> The training details are provided in Section 4.4. We agree that our approach is still a kind of BC. However, the novelty of Auto-UI lies within (i) a multimodal agent for autonomous UI control that can directly interact with the screens, thus circumventing the constraints of environment parsing and application-specific API access, and (ii) a chain-of-action technique that leverages the previously executed actions and future action plans to help the agent decide what action to execute at each step. Therefore, our work represents a significant departure from prior studies in the field. In addition, Auto-UI achieves state-of-the-art performance with an action type prediction accuracy of 90% and an action success rate of 74%. Notably, Auto-UI can infer an action as fast as within less than one second.
>
> > W2: About the ICL baseline.
>
> Firstly, the ICL baseline is implemented following the prior study [1]. To help readers understand the details, we have exactly provided examples of the ICL baseline in Appendix A.3 with represented actions and OCRed results.
>
> Secondly, we have conducted additional analysis on the action plan level. The result is shown below.
>
> | Model                     | Overall | Action Type  | Click        | Scroll       |
> | ------------------------- | ------- | ------- | ------- | ------- |
> | ChatGPT | 5.93    | 41.72       | 8.50  | 4.00   |
> | Auto-UI | 68.24   | 87.03       | 58.34 | 82.74  |
>
> Indeed, we see that the ICL method (ChatGPT) is quite accurate at predicting the action type (41.72%) but fails at lower-level executions, e.g., clicking positions (8.5%) and scrolling directions (4.0%).
>
> Thirdly, this is exactly the motivation of our work. Previous work commonly uses HTML syntax while we argue that parsing the visual environment into textual elements may be prone to error propagation or information loss and the parsed elements generate lengthy inputs, thus leading to inference inefficiency. Following the reviewer’s comment, we add the results of a strong multimodal ICL baseline (GPT-4V). The results show that our proposed approach is still much better than the baseline.
>
> | Model                     | Overall | General  |  Install  | GoogleApps  | Single  |
> | ------------------------- | ------- | ------- | ------- | ------- | ------- |
> | GPT-4V  | 52.96   | 43.01   | 46.14   | 49.18      | 78.29  |
> | Auto-UI | 74.27   | 68.24   | 76.89   | 71.37      | 84.58  |
>
> > W3: “Similar to above, the third baseline, fine-tuning LLMs, need to have a version with multimodal inputs.”
>
> Yes. We have added the results of fine-tuned LLMs with multimodal inputs (LLaVA) [1]. This can be seen as a variant of our framework by changing the backbone modules. We see that LLaVA is able to achieve slightly better performance gains and our proposed chain-of-action technique is also effective at enhancing the performance further.
>
> | Model                     | General  |
> | ------------------------- | ------- |
> | T5 backbone            | 68.24 |
> | LLaVA  backbone      | 68.70   |
> | w/o chain of actions | 58.40   |
>
> > W4 & Q2: “An error analysis is required both on the quantitative and qualitative sides, what are the major errors that these models exhibit?”
>
> We have conducted error analysis in Section 5.1 and find that the major errors lie within the click region and scroll direction predictions. Although the model is able to predict the right action most of the time, it tends to click the wrong place or scroll in the wrong direction. The result reveals a future direction of improving the model’s ability to understand the screen layouts, e.g., using more advanced vision features.
>
> > Q1: “I’m a bit surprised that the language decoder is able to predict tokens as precisely as four decimal places, or is the actual precision here not important? I.e., could you not simply split image screens into patches and just use their centers as the coordinate representations? (And the more patches you grid the screen, the more precise it would be.)”
>
> Yes. Our results show that the actual precision here is not important. For clicking actions, as defined in [2], a click action is considered correct if its touch point and lift point fall within a 14% screen distance from the gold gestures or occur within the same detected bounding box with the gold gestures. For scrolling actions, a scroll action is considered correct if it has the same scroll axis as the gold gesture.
>
> Actually, we did not use patches but the global (pooled) representation of the vision features as described in Section 3.2 (Encoding).
>
> [1] Haotian Liu, Chunyuan Li, Qingyang Wu, and Yong Jae Lee. Visual instruction tuning. NeurIPS 2023.
> [2] Christopher Rawles, Alice Li, Daniel Rodriguez, Oriana Riva, and Timothy Lillicrap. Android in the wild: A large-scale dataset for Android device control. arXiv preprint arXiv:2307.10088, 2023.

---

> > ### Comment · Reviewer_CMSv · 2023-11-20
> >
> > Thanks for the efforts to provide the above additional results, I do appreciate them.
> >
> > While the numbers seem decent, my main concerns still stand, which are exactly what the two novelties mentioned in your response.
> > For the first novelty mentioned, that is merely a design choice of adapting modules to the correct corresponding targeted domains.
> >
> > The second novelty claimed, if at all, is the main issue -- it is not really novel, at least not as much as it is claimed.
> > The chain-of-action, is essentially just concatenating action history, which is very straightforward to come up with when approaching problems like this where a series of actions are to be taken.
> > Furthermore, it is not taking the image history into consideration, which can potentially bring benefits and enrich the technical parts of the work (which also draws an analogy to state-action pairs of a multimodal RL agent).
> > Therefore, this being claimed throughout the paper as the main novelty, does not really warrant the level of technical novelty contribution in learning conferences such as ICLR.
> >
> > I think this work has a chance to make a nice case in conferences like CSCW and/or NAACL, I encourage the authors to submit the manuscript to those types of venues.

---

> ### Author Response · Authors · 2023-11-21
>
> Thanks for the further comment. We understand your concerns on the novelty. Though we respect the thoughtful comments, we humbly think those concerns are caused by misunderstanding, which we will explain in detail below.
>
> > “For the first novelty mentioned, that is merely a design choice of adapting modules to the correct corresponding targeted domains.”
>
> To the best of our knowledge, our work is the first to present a multimodal language agent framework for challenging autonomous user interface control.  The novelty of our approach can be distilled into the following two key facets:
>
> **(i) Addressing a non-trivial problem**: Our focus on developing multimodal autonomous language agents represents a cutting-edge research direction that has garnered significant interest. However, the inherent challenge lies in devising a comprehensive solution for multimodal perception and reasoning, encompassing aspects such as planning, memory, and decision-making. Existing studies have, to date, grappled with a sandbox setting, relying on external tools and application-specific APIs for environment perception and action interpretation. Regrettably, these approaches have left unresolved challenges in their wake.
>
> **(ii) Pioneering a new paradigm**: To address the complexities outlined above, our work introduces Auto-UI, a simple, effective, and efficient solution grounded in first principles thinking. This framework bypasses the need for intricate environment parsing or dependence on application-specific APIs. Auto-UI, propelled by our proposed mechanisms, achieves a state-of-the-art performance. As echoed by Reviewer BNVX and Reviewer tvGC, our work is an effective solution for real-world applications of autonomous agents.
>
> Consequently, characterizing our work as "merely a design choice of adapting modules to the correct corresponding targeted domains" oversimplifies its innovative contributions. Rather, it signifies a more efficient and reliable approach to task automation (echoed by Reviewer BNVX). Even so, proposing a design choice of adapting modules to the correct corresponding targeted domains has also been widely accepted in existing publications [1-3]. In light of these considerations, we believe our work holds significant value for the ICLR community.
>
> > About the novelty of the chain-of-action technique.
>
> Firstly, we wish to address a potential misunderstanding regarding the chain-of-action approach. It is NOT merely "concatenating action history"; rather, it consists of two crucial parts: a chain of previous action histories on the input side and a chain of future action plans on the output side. This innovative approach encourages the agent to utilize action history and formulate future plans prior to predicting the current action. In essence, it mirrors the memory and planning mechanisms observed in language agents.
>
> Secondly, while we did contemplate incorporating image history, our empirical findings indicated that its inclusion did not result in performance improvements. Consequently, we opted for the current implementation to maintain simplicity rather than pursuing unnecessary complexity.
>
> Thirdly, we’d like to humbly share [a recent article by Michael Black](https://perceiving-systems.blog/en/post/novelty-in-science) on novelty in research: simplicity is not in conflict with novelty. We humbly believe that a simple method, like Auto-UI, that solves an important problem but no one thought about before, is novel. We appreciate your thinking alike.
>
> In conclusion, we hope our clarifications can alleviate your concerns  and you can consider our work more favorably.
>
> *References:*
>
> [1] Yehudai, Asaf, Matan Vetzler, Yosi Mass, Koren Lazar, Doron Cohen, and Boaz Carmeli. QAID: Question Answering Inspired Few-shot Intent Detection. ICLR 2023.
>
> [2] Chirkova, Nadezhda, and Sergey Troshin. CodeBPE: Investigating Subtokenization Options for Large Language Model Pretraining on Source Code. ICLR 2023.
>
> [3] Robinson, Joshua, and David Wingate. Leveraging Large Language Models for Multiple Choice Question Answering. ICLR 2023.
>
> Thanks,
> Authors

---

### Official Review · Reviewer_tvGC · 2023-10-30

**Soundness:** 3 good
**Presentation:** 2 fair
**Contribution:** 2 fair
**Rating:** 5
**Confidence:** 4

**Summary:**

This paper proposes an autonomous UI agent called Auto-UI that can interact in a multimodal UI environment without environment parsing or application-dependent API access. Specifically, it proposes a chain-of-action technique to help the agent make decisions.

**Strengths:**

1. It is novel that the paper pays attention to the limitations in the real-world applications of autonomous agents and seeks to provide an agent that does not need extra intermediate environment parsing or interval application-dependent APIs.

2. The paper proposes a chain-of-action technique which helps the agent to decide step-by-step.

**Weaknesses:**

1. The Figure 1 in this paper is somewhat not clear enough, making it difficult to understand the two paradigms in (a) and (b).

2. The author does not provide a specific explanation of the Sandbox Paradigm and the First Principles Thinking Paradigm, which is confused.

3. We find some grammar mistakes in the paper, for example, on page 2, paragraph 2, line 5, do you want to express inefficiency instead of efficiency?

4. The authors don't explain exactly what touch_point, lift_point, etc. mean in the first place, causing some confusion.

5. The authors do not provide a specific example between Auto UI and other baselines in Section 5, which is not clear to understand the effectiveness of the provided Auto UI.

**Questions:**

In Section 4.3, why do you use 14% instead of other number to evaluate the correction of a click action, could you provide some references?

---

> ### Author Response · Authors · 2023-11-19
>
> Thanks for your insightful review and constructive feedback.
>
> > W1 & W2 “The Figure 1 in this paper is somewhat not clear enough, making it difficult to understand the two paradigms in (a) and (b).” & “a specific explanation of the Sandbox Paradigm and the First Principles Thinking Paradigm”
>
> We provide a detailed explanation of the two paradigms from the third and fifth paragraphs in the introduction. We elaborate on it as follows:
>
> *(a) sandbox paradigm*: depends on the intermediate transformation between environments and agents
>
> - input side: relies on external tools such as optical character recognition (OCR) and icon detectors to parse the environment into textual elements
>
> - output side: requires accessing internal APIs to interact with the environment, e.g., using a JavaScript element selection on a webpage or a Python interpreter to execute actions.
>
> *(b) first principles thinking paradigm*: allows direct interactions on the screen without needing access to intermediate environment parsing or interval application-dependent APIs.
>
> > W3: “We find some grammar mistakes in the paper, for example, on page 2, paragraph 2, line 5, do you want to express inefficiency instead of efficiency?”
>
> Yes. It is inefficiency. We have fixed the typo in the revised version.
>
> > W4: “The authors don't explain exactly what touch_point, lift_point, etc. mean in the first place, causing some confusion.”
>
> Touch_point and lift_point simply mean the axis of touching and lifting. As the notation is widely accepted in related studies and we believe it is also commonsense, we provide examples when discussing coordinate normalization in Section 3.3. Even so, following the reviewer’s comment, we have added an explanation in the first place (Figure 1).
>
> > W5: A specific example between Auto UI and other baselines in Section 5.
>
> Section 5 is about the analysis of our approach, which follows the same model architecture. We believe the reviewer means Section 4 (please correct us if it is wrong). We have provided a detailed description of the baselines in Section 4.2. Actually, the example of the specialized UI agent and fine-tuned LLMs is illustrated in Figure 1 while the example of the in-context learning LLMs is provided in Appendix A3.
>
> > Q: “In Section 4.3, why do you use 14% instead of other number to evaluate the correction of a click action, could you provide some references?”
>
> The choice basically follows the previous study [1] which releases the dataset and defines the evaluation criteria. We follow the setting for a fair comparison.
>
> [1] Christopher Rawles, Alice Li, Daniel Rodriguez, Oriana Riva, and Timothy Lillicrap. Android in the wild: A large-scale dataset for Android device control. arXiv preprint arXiv:2307.10088, 2023.

---

### Official Review · Reviewer_y4jm · 2023-11-02

**Soundness:** 3 good
**Presentation:** 3 good
**Contribution:** 2 fair
**Rating:** 5
**Confidence:** 3

**Summary:**

This paper presents a multimodal work for Auto-UI, it proposes to leverage the chain-of-action (including previous history actions and future actions) for model prediction. Their model builds on the top of Llama 2 with an image encoder (for screen image). Empirical experiments on the AITW dataset shows very promising results.

**Strengths:**

1. This work proposes a chain of action operation, leveraging the action history and future actions for current action prediction.
2. Based on Llama 2, it incorporates a pretrained image encoder into the pretrained LLM for action decision, and shows promising results on AITW dataset.

**Weaknesses:**

1. A potential weakness is where is the gain from? It looks PaLM and ChatGPT are pretty low on this dataset, while they only take text input, and BC models and Auto-UI models take image screen as input, and get very high results, it is unclear where is the gain from? image encoder? or a chain of action input?

**Questions:**

I try to understand the setting of the experiments, and why the strong PaLM and ChatGPT baselines are so low. Based on the main Table 2, it looks the most gain is from the image encoder, right? Since PaLM-CoT and ChatGPT-CoT only take text input, and their performance is pretty low, and also similarly for Llama 2. Is this right? Probably needs a baseline/ablation to see the performance of model without image encoder.

---

> ### Author Response · Authors · 2023-11-19
>
> Thanks for your insightful review and constructive feedback.
>
> > W & Q: Where is the gain from?
>
> The performance gain mainly comes from two aspects, multimodal perception and chain-of-action technique. Following the reviewer’s suggestion, we have conducted additional experiments to ablate the performance of the model without the image encoder (w/o multimodal perception) but use the parsed HTML layout instead.
>
> | Model                     | Overall |
> | ------------------------- | ------- |
> | Auto-UI                   | 74.27   |
> | w/o multimodal perception |   62.70      |
> | w/o chain of actions      | 68.53   |
>
> The key takeaways are:
>
> (i) Multimodal perception is critical compared with existing baseline language agents built purely on large language models. The result shows that our model out of first principles thinking can serve as a strong autonomous agent.
>
> (ii) The chain-of-action technique helps improve the ability of our multimodal agent further, by leveraging the previously executed actions and future action plans to help the agent decide what action to execute at each step. As a result, the memory and planning ability can be enhanced.
>
> It is reasonable that the baseline results are low due to the lack of vision encoders. As a result, they may suffer from error propagation or information loss (as discussed in the introduction section). To have a deeper understanding of the performance, we have conducted additional analysis on the performance regarding the action type and action execution (i.e., clicking accurate positions and scrolling in the correct direction).
>
> | Model                     | Overall | Action Type  | Click        | Scroll       |
> | ------------------------- | ------- | ------- | ------- | ------- |
> | ChatGPT | 5.93    | 41.72       | 8.50  | 4.00   |
> | Auto-UI | 68.24   | 87.03       | 58.34 | 82.74  |
>
> We find that the ICL method (ChatGPT) is quite accurate at predicting the action type (41.72%) but fails at lower-level executions, e.g., clicking positions (8.5%) and scrolling directions (4.0%).

---

### Meta-Review · Area_Chair_c9CR · 2023-12-08

**Metareview:**

The paper proposes an approach that directly takes visual input (screenshots) of a UI as well as language command as the input, without asking the environment to parse the screen. The approach also proposes "chain of actions" that essentially sends previous actions and a plan of future actions (generated by an LLM) to the model to predict the current action. I like the overall approach, although none of these aspects is significantly novel. The way the action history and plan is represented would only be valid when the interaction remains on the same screen, because coordinates of touch point alone do not carry enough information about an action. It is also becoming a common place to take visual input directly, without using parsed UI meta data, for interaction agents. I was excited when seeing Figure 1 which details components of the agent. However, the figure is simply there as a conceptual explanation that doesn't technically connect to the model architecture discussed later.

The reviewers are on the fence. On one hand the reviewers think the paper is addressing an important problem and the overall approach is interesting. On the other hand, there is a lack of a strong voice to vouch for the paper. Both Reviewer BNVX and Reviewer CMSv questioned the novelty of chain of actions, and pointed out it is "essentially just concatenating action history" or "primarily concatenate the output actions". The reviewers pointed out a number of presentation issues that need to be clarified. The paper also seems to make big claims about simplification with the agent IO and leveraging action history. Overall, the novelty of the work is limited.

**Justification For Why Not Higher Score:**

The paper tackles an important problem and investigates an interesting setup.

**Justification For Why Not Lower Score:**

The work offers limited technical novelties.

---

### Decision · Program_Chairs · 2024-01-16

Reject